# Female-Male and Female-Female Social Interactions of Captive Kept Capercaillie (*Tetrao Urogallus*) and Its Consequences in Planning Breeding Programs

**DOI:** 10.3390/ani10040583

**Published:** 2020-03-30

**Authors:** Joanna Rosenberger, Artur Kowalczyk, Ewa Łukaszewicz, Tomasz Strzała

**Affiliations:** 1Institute of Animal Breeding, Division of Poultry Breeding, Wroclaw University of Environmental and Life Sciences, 50-375 Wrocław, Poland; 2Department of Genetics, Wroclaw University of Environmental and Life Sciences, 50-375 Wrocław, Poland

**Keywords:** capercaillie, captivity, breeding behavior, tooting, mate selection, aggression, nest site competition

## Abstract

**Simple Summary:**

The Capercaillie is one of the most endangered bird species in many European countries. To prevent further population decline, breeding centers where birds are bred and later released into the wild were established. However, in many Capercaillie breeding stations, reproductive success is limited due to low fertility and problems with incubation behavior by females. Reasons for these problems are usually multidimensional, including misunderstanding Capercaillie behavior in the breeding environment. Research was conducted in Capercaillie breeding centers where birds were monitored 24 h/day by cameras. We observed that females preferred to mate with males with a longer tooting activity, but when males became too insistent and started to chase the females, they avoided contact. Even when the density of females was only one bird per 132 m^2^, nesting site competition occurred. In 67% of the nests, female intruder’s presence could be observed. Interactions between females were mainly antagonistic, and while none of the birds were harmed, this caused egg damage and nest abandonment. Nesting site competition in captive Capercaillie is high and may lead to antagonistic interaction between females, consequently lowering breeding success. Our research showed also that mate preference of males by females could be predicted by observing the male’s tooting activity.

**Abstract:**

Capercaillie behavior, both in the wild and in captivity, is poorly known due to this species’ secretive way of life. Female-male and female–female social organization and interactions are especially poorly documented. The research was conducted in Capercaillie Breeding Center in Wisła Forestry District where a breeding flock is kept throughout the year. Thanks to video monitoring, we were able to observe mate choice, and then later, female–female interactions during laying and incubation period. Male individual variation in tooting latency and duration were recorded. Females’ interest in males was related to males’ tooting activity, but when males became too insistent and started to chase the females, the females avoided contact with them. There was a significant relationship between calendar date and when tooting starts, and between the tooting duration the female spent with a male. Two incidents of female-male aggression caused by competition for food were observed. Female intruder presence and competition for nesting place was observed in 66.67% nests. Most female–female interactions were limited to threat posturing, but fights and attempts to push out the intruder from the nest occurred as well. Such interactions may lead to nest abandonment and egg destruction, lowering the breeding success.

## 1. Introduction

A main aim of endangered species captive breeding centers, such as at zoological gardens, is to propagate progeny to be released into the wild [1,2]. One such species is Capercaillie (*Tetrao urogallus*), where each individual is valuable and any factor reducing reproductive success should be avoided. To increase breeding success, assisted reproduction methods such as artificial insemination [3,4] can be used. This is best accompanied by close observation of in-situ breeding behavior which can help to predict successful mating and fertilization, including identifying the father of offspring. Another common method to increase fecundity is to remove and artificially incubate eggs to stimulate continuous laying or production of a second clutch [3,5]. Preventing egg damage in parent-incubated eggs is also crucial. Eggs may be destroyed unintentionally by parents (thin eggshell breaks during laying, incubation, or during fight with other animals) or intentionally (by another individual, parent, or predator). One example is the egg eating problem described mainly in poultry species [6], but also occurs in non-domesticated bird species [7,8]. Abnormal behavior, including filial cannibalism and other disorders, is unfortunately common in birds kept in captivity [9,10,11] and can be reduced by suitable management practices such as behavioral enrichment and ample living space. Careful animal observation is necessary to identify normal and abnormal or problematic behaviors.

Many research studies recall examples of pathological behavior in Capercaillies, indicating that this species is particularly susceptible to behavioral disorders. Abnormal behavior was described in males showing increased aggression towards other species, including humans, not reaching the lek areas and displaying tooting behavior outside the lek. Abnormally behaving females avoided males, appear to be tamed, and took acceptance to mate position in front of humans [12,13,14]. Cited authors suspect those behaviors may be caused by disorders of testosterone levels in both sexes. Despite research carried out on this topic, little is known about other behavioral disorders in Capercaillie, especially of birds kept in captivity.

The Capercaillies secretive way of life makes it hard to study their behavior. For most of the year they are solitary birds, hard to track and observe in their natural habitat. Small populations in many European countries [13,15,16] makes observation additionally troublesome because of limited availability of observation places and legal restrictions that were made for the protection of birds. While some research has been conducted at lekking areas [17,18,19] where birds gather, usually annually, there is almost nothing known about social interactions outside leks. Females meet males at lek, where after choosing a partner, display their readiness to mate by assuming an acceptance position: lowering their bodies to the ground and slightly spreading their wings. Wegge and Rolstad [19] and Wegge and Larsen [17] indicate that 95% of the Capercaillie females visit only one lek in the season, however if the lek is occupied by a small number of males, females more readily move between several leks [18]. Some research was conducted on Capercaillie incubation rhythm [20], but almost nothing is known about female–female interactions during nesting season. Despite the fact that such meetings may occur as suggested by Storch and Segelbacher [21] observations of clutches of two females in one nest. Authors based on nest analysis not witnessing the interactions themselves. When Capercaillie females are laying and incubating they are very sensitive to human disturbance [22,23], which can lead to nesting failure either directly (abandonment of nests) [24] or indirectly (by alerting avian predators to the nest location) [25,26,27]. All of these factors make Capercaillies nests in the wild hard to find and observe.

Birds kept in captivity provide study opportunities, but many important aspects have been overlooked, such as male activity and how it changes during breading season, female mate choice, and interactions between individuals during nesting. Especially little is known about social interactions between females and nesting site competition in captivity. We assume that looking at bird behavior in the breeding centers, where conditions are similar to natural but individuals are kept in higher density, could allow us to suppose how interactions in wild may proceed. Made observations may improve our understanding of Capercaillie breading behavior in captive kept individuals, but it may also improve us with knowledge of behavioral patterns of their wild counterparts. Simultaneously conducted research is essential in understanding and improving the conditions of Capercaillie in breeding centers and zoological gardens, as well as avoiding reduced breeding success.

## 2. Materials and Methods

### 2.1. Maintenance Conditions

Research was conducted at the Silesian Beskid in Capercaillie Breeding Centre Wisła Forestry District (CBC-WFD) (49°32′05.4” N, 18°55′58.1” E), away from human settlement. The breeding flock that consisted annually of 13–15 males and 30–40 females, is kept throughout the year in wooden, roofed aviaries that are divided into sub-units. The aviaries where observations were made consisted of six small sub-units with dimensions of 6.1 × 4.0 m and a height of 2.5 m. The sub-units were equipped with a perch and four pines (*Pinus sylvestris*) or spruces (*Picea abies*) replaced by fresh material after drying. The floor was covered with sand, which was screened daily. Beside small sub-units available for birds for the entire year, aviaries had six small outside areas 4 × 4.5 m and a height of 4 m poured with sand and four large fenced yards (10 × 20 m, height of 5 m) that was part of forest with elements of the natural habitat (trees, shrubs, grass, windthrows). For each female there was space 131.8 m^2^, while for each male: 25.5 m^2^.

Flock was divided into two “family groups” (Aviary A1 and A2) that consists two males in each (A1: 119 and 120; A2: 105 and 122) and four females in each (aged one to ten years). For each family group three small sub-units were available: left and right only for one of the males, middle only for the females. Atmospheric and lighting conditions were identical for both family groups, and both outside aviaries faced south, which provided plentiful sunshine. Hens were able to move freely between sub-units inside aviary A1 or A2 thanks to holes with a diameter of 17–18 cm in walls, while the larger males movements are restricted and stay in their sub-unit. Females have freedom in choosing a nesting place: inside aviary or in the yard.

Birds were fed once a day with poultry mash, live crickets, fruits (blueberries, cranberries) and fresh buds of deciduous trees. Needles from pine and spruce trees placed inside aviaries were an additional source of food. Water acidified with lemon juice (to prevent excessive development of gastrointestinal microbiota in the cecum, and in consequence, inflammation) was changed daily. To reduce risk of disease, food and water vessels were daily washed and sterilized at 200 °C. To prevent birds from eating food that might have been contaminated during contact with the ground, nourishment was placed in bowls that were hanged above boxes with mesh. All food and water was provided by bird keepers that Capercaillies were accustomed to.

### 2.2. Making Observations

Extensive monitoring was installed at the aviaries in the beginning of the March: three cameras recorded birds behavior in each sub-unit where females visit males, while the rest of the cameras (one camera per nest) were placed near the nests to monitor behavior during incubation. The behavior of four males (male 105 was eight years old, male 119 two years, male 120 three years, and male 122 one year) and their interactions with females were monitored from April 15 to May 12 in one year. We recorded the duration of female presence in male sub-units during the day, male tooting latency (time when male started tooting song) and daily tooting duration. Behaviors were considered as tooting only when the male issued a toot song and took a tooting posture with raised, spread tail and lowered wings. Interactions were categorized as: ignoring same sex, ignoring opposite sex, and sex specific behaviors of males (tooting, chasing female), and females (observe male tooting, interest in male, or escape from male) (Table 1).

Recording of female–female interactions directly in the nest and close to the nest area (up to 50 cm from nests) were made at 18 nests over four years of observations (Table 1). Nest monitoring was made starting during the laying phase, after nests were found by bird’s keepers (2–14 days, mean 4.5, before incubation started;) and during the entire incubation period (25–27 days). All females were individually marked with rings, one number and one color ring.

All recordings were made 24 h per day, and birds were accustomed to cameras and did not react to their view or sound. Videos of males were continuous (648 h of each male). Cameras near female nests started to save data 5 s before movement detection.

### 2.3. Determination of Paternity

To confirm the behavioral observations connected with the mating process we used genetic analyses to reveal paternity of offspring of observed birds. We matched behavioral observations of observed individuals with their mating processes and paternity. Blood samples were taken from every adult and offspring in the breeding centre. Two people were involved in this procedure: one caught the bird, the other after raising the wing was disinfecting the place of blood collection (preparation with Octeniderm^®^ Schülke, Germany, then a single puncture (with a disposable needle with a diameter of 0.45 mm) in ulnar vein (*v. ulnariscutanea*) was made. A fragment (up to 5 cm^2^) of filter paper was applied to the drop of blood that appeared. The blood stain required for genetic analysis did not exceed 1 cm^2^. The individual was released into the aviary after the bleeding had stopped.

DNA was isolated from blood stains with Sherlock AX (A&A Biotechnology) according to producer’s instruction. After isolation, DNA quality and quantity were assessed with Nanodrop (Thermo Scientific). Next, after DNA dilution to uniform samples concentration, 29 microsatellite loci for all individuals (Table 2) as a set of nine multiplex reactions with Qiagen Multiplex PCR kit (in 10 µL volume) were amplified. The lengths of PCR product lengths were read with Applied Biosystems 3730XL gene analyzer and GeneMarker 2.4.2 SoftGenetics, PA, USA (available at www.softgenetics.com).

Parental analyses were conducted manually. We set all the possible parents along with the offspring and looked for alleles that did not match between parent–offspring pairs. After we excluded all potential fathers/mothers except one, we confirmed offspring paternity for the last father/mother set with all loci.

### 2.4. Statistics

Data were analyzed using Minitab^®^ 17.1.0, Minitab Inc., State College, PA, USA. Our data did not have the normal distribution (tested with t-distribution) so non-parametric statistical tests were used. Correlation between daily time females spent with males in their sub-units and tooting duration as well as correlation between female presence in male sub-unit at night and tooting duration were tested with two-tailed Wilcoxon test. Male factor’s influence on tooting latency, tooting duration, and female presence, as well as date’s influence on tooting latency, tooting duration, and duration of female presence with male and time spent with male daily relation with female readiness for mating were tested with the Kruskal–Wallis test. The Kruskal–Wallis test was also used to test differences in tooting latency, tooting duration and time females spent with males depending on the month (April, May). Pearson’s Chi-square test was conducted to investigate the relationship between male and female behavior. All tests were conducted on the confidence level 95%.

### 2.5. Ethical Note

The National Forestry in Wisła District has the permission (DOP-OZGIZ.6401.03.171.2011.km, dated on: May 10, 2011; expiry date: December 31, 2021) for keeping, reproduction, and collection of the biological materials for experimental purposes of adult and juvenile birds in the Capercaillie Breeding Center in Wisła Forestry District, Poland. The permission was issued by the General Director of Environmental Protection and signed by Michał Kiełsznia for keeping.

Blood sampling procedure was approved by the II Local Ethics Commission for Experiments Carried on Animals (Permit: No. 13/2017).

## 3. Results

### 3.1. Male/Female Interactions

A significant positive correlation was observed between the time the female spent with a male daily and duration of male tooting activity (*p* < 0.001; z-value = −6.643) (Figure 1). Female presence at night in the male sub-unit was positively correlated with duration of male tooting (*p* < 0.001; z-value = −8.174). Latency of tooting was dependent on male individual factor (*p* < 0.001; DF = 3; H = 40.70). Older males (105 and 120) started their activity earlier (z = −3.37; z = −3.04, respectively) than younger (122 and 119) (z = 3.45; z = 4.65, respectively) (Table 3). Duration of tooting was also dependent on male individual factor (*p* < 0.001; DF = 3; H = 71.74). Males 105 and 120 were tooting longer (z = 5.78; z = 3.46, respectively) than males 119 and 122 (z = −2.68; z = −6.56, respectively). Females had strong preferences in spending time with a particular male (*p* < 0.001; DF = 3; H = 52.43), choosing male 105 (z = 5.68) and 120 (z = 1.78) instead of male 119 (z = −1.91) and 122 (z = −5.55). No significant correlation was stated between the date of observation and tooting duration (p = 0.381; DF = 76; H = 79.13), the date and time the female spent with a male in their sub-unit (*p* = 0.271; DF = 80; H = 87.27) or the date and tooting latency (*p* = 0.288; DF = 67; H = 72.99). The relationship between duration of tooting and month of observation was detected (*p* = 0.002; DF = 1; H = 8.87). In April, males were tooting longer (z = 2.98) than in May (z = −2.98). No relationship between the month and tooting latency (*p* = 0.276; DF = 1; H = 1.19) or the time females spent with males (*p* = 0.563; DF = 1; H = 0.34) was noticed.

Only in male 105 sub-unit, females’ readiness to mate (15th–20th and 23rd–27th April) and mating (24th and 25th April) were observed. Female readiness to mate was not correlated with the time females spent with a male during the observation period (*p* = 1.000; DF = 80; H = 23.10), neither when the analyzed data were limited only to days when females were showing readiness to mate (*p* = 0.943; DF = 29; H = 18.04). The genetic analysis showed that the father of all chicks in A1 was male 119 and in A2 male 105. With the unexpected finding that females chose to mate with an individual with less tooting activity and a male they spent less time with, we analyzed the change of males’ behavior between April and May, and noticed that male 119 tooting duration (*p* = 0.341; DF = 1; H =0.91) and time females spent with a male (*p* = 0.164; DF = 1; H =1.94) did not differ significantly between months, while in the case of male 120, tooting duration (*p* = 0.006; DF = 1; H = 7.50) and time females spent with a male (*p* = 0.011; DF = 1; H = 6.52) were dependent on the month. In April, male 120 tooting duration and time females spent with a male were lower compared to May (April z = 2.74; May z = −2.74 and April z = 2.55; May z = −2.55, respectively). The comparison of male 119 and 120 activity in April and May indicated that the decrease of tooting activity of male 120 was greater than for male 119 (Figure 2) and the decrease of time females spent with a male was seen in male 120, while in the case of male 119, females were spending slightly more time with that male in May than in April (Figure 3).

Female and male behavior was correlated χ2 (DF = 9, N = 272) = 185.227. Females were interested in males (observation of male activity or female readiness to mate) only when males were tooting, but when males became more insistent and started to chase the females, the females were avoiding contact and escaped. Out of all observed tooting activities, females were ignoring males in 41.77%, avoiding contact in 22.15%, observing males in 20.89% and showing readiness to mate in 15.19% of cases. Most often the chase of the female in relation to all behaviors presented by the males was done by male 122 (36.84%) and 119 (26.56%), and least often by male 105 (11.00%) and 120 (15.73%).

### 3.2. Female/Male Aggression

We observed two cases of aggressive behavior between female and male 105, on May 10 and 11. In the first case, the female went down from the perch and stopped near the feeding place with fresh leaves and buds. The male was laying nearby, while the female started to eat and then lied down. When male 105 moved, female spread the tail and started to show aggressive behavior. The male was not showing any interest in the female, nor showed any signs of antagonistic behavior, but the female approached the male and began to peck him. The male spread his tail and started to walk toward the female while hen was still pecking and moving back. The female circled the male standing behind him, still with the spread tail. When the male turned toward her, the female fled and hid behind the trees.

The next day aggressive behavior continued between the same male and female. Like the day before, antagonistic behavior occurred near the place where birds had access to leaves and buds. The male walked toward the laying female with a spread tail but was not tooting. Female stood up and tried to outrun the male by stretching her neck and making aggressive pecks in the males direction. After two minutes, she spread her tail and shortly after this, she walked away and left the male sub-unit. All that time, the male was walking toward the female with a spread tail and raised neck, not showing any further aggressive behavior.

### 3.3. Female/Female Antagonistic Interactions During Laying and Incubation Period

During four years of observation and tracing the fate of 18 nests, in 12 nests we witnessed 99 interactions in distance about 50 cm from the nest. Observed behaviors during female interaction were mostly antagonistic (Table 4). We used five categories of observed reactions: any reaction of another female presence near the nest (NR); threatening attitude with a raised head, ruffled feathers on throat and making sounds (TA) (Figure 4); attack of one bird on another (AT) (Figure 5); fight over nest using a beak (NF), and attempts of pressing to push a female out from the nest (PN). No injuries of females caused by antagonistic behavior were observed. NR was observed when non breeding females were foraging near the nest place or were exploring nest site but were not interested in the nest itself. One observed behavior may have smoothly transitioned into another. In the case when NR turns into TA, it was observed when non nesting females stayed near the nest area for longer than 5 and 4 min.

We did not find egg crushing when antagonistic behavior was limited to the TA, but when females were more aggressive, three cases of egg destroying occurred. One egg was crushed during a fight between females, one egg was removed from the nest by a parent and later broken, the last one was eaten by an intruder.

In total, eight cases in five nests were observed when another female started incubating when the nesting female made a nest recess. Female intruders showed a characteristic behavior: long-term displacement of eggs, nest material, awkward nesting, not covering all eggs while incubating and nervous behavior during incubation. The most destructive after-effects were observed when the intruder female took over one of the nests: firstly, the female that the nest belonged to was trying to retake the nest for two days only returning to the nesting area but not taking any other actions, however this was followed later by more serious aggression. After unsuccessful attempts of pushing the intruder from the nest, the birds started to fight. The intruder did not leave the nest and nesting female sat near the nest and took one of the eggs from the nest. As a consequence of those actions, two eggs were destroyed during the fight, and another one was eaten by the intruder. When the intruder left, the nesting female sat back on the eggs. In another nest, the nesting female succeeded in chasing away the intruder by pushing and pecking. Two other nests were taken by the intruder female, one of the displaced females re-nested, one did not. Another case of nest abandonment before incubation occurred when two females started laying into the same nest but neither started the incubation, so the eggs were taken to an artificial incubator. In the three remaining nests where the intruder incubation occurred, the intruder left the nest when the nesting female returned.

## 4. Discussion

### 4.1. Female/Male Interactions

There are different mating systems among birds: monogamy is the most common in most species [31], but among certain groups, including grouse, polygamy is dominant [32]. Choosing the best mate is crucial for all the females because offspring fitness depends highly on that choice. Mate choice can be based on many factors like the size of the occupied territory and its abundance of resources, appearance, or behavior [33,34,35,36], and the significance of these factors varies for different species. According to Storch [18], two-thirds of Capercaillie females visit more than one lek, but the others remain faithful to one, many times over the years. In the wild, the time of day affected female presence at the lek area [18], while in our captive population, the time of day was not significantly correlated with it. In Wegge and Rolstad’s [19] observations, the females were absent at the lek areas at night and arrive early in the morning. Limited living area may account for our observed differences. It is worth noting that females were spending the night in a particular male’s sub-unit instead of their own. This might be because hens in the wild need a secure roosting place, while in captivity, all sub-units are the same and safe from predators, so they can afford to stay with a male to observe all their activities.

When females arrive to the lek, they need to make mate choices. In the case of Capercaillie, usually the dominant male is preferred [13] but what makes females choose a particular male is not well known. Our research from the breeding center indicates that the females do not always choose the oldest male (unpublished observations). It seems that the age of the male may be one of the female criteria for the assessment of the male, but not the only one. One-year-old males are physiologically able to mate with females because they are producing good quality semen and showing attempts to copulate [37], but they are probably not able to compete with older individuals in the wild and females are not interested in them both in the wild and in captivity. During our observations, females spent more time with older males (male 105 over 122 and male 120 over 119) that were tooting longer. Among older males, females showed clearly greater interest in male that was eight years old (male 105 over 120), spending more time with him and showing readiness for mating. While females’ preferences of males were clear in A2 and behavioral observations were consistent with the results of paternity of chicks, in A1 the father of all chicks was male 119, that was one year younger than second male that females had access to. Even though females spend more time with the male 120, especially at the beginning of observations, they did not mate with him.

We did not witness any female showing acceptance of the position for mating in A1, also the eggs laid in April and the first half of May were unfertilized. Only eggs from the second half of May were fertilized, so successful mating probably occurred after our observations. Tooting activity changed over April and May: while male 120 was tooting less in May compared to April, tooting activity of male 119 did not change. We suppose it could have been a factor that influenced the females’ choice. Our observations indicate that in the case of Capercaillie, the choice of the male depends on many factors, and probably both male age and tooting activity are involved. We also noted that the male’s reaction to the hens’ appearance is also a factor. The issue of mate choice by Capercaillie females can be a challenging in breeding centers, because hens have no options to move to a different lek as in wild birds, and subsequently, may lay unfertilized eggs. Wild Capercaillie leks are occupied by several males, and [17,18,19] it seems to be important to provide females with the widest possible choice.

Many animal females may avoid interactions with aggressive or overzealous males as they could be at risk of injury [38,39]. This was also supported by our observations. In previous studies, some males showed aggression toward female dummies intended for semen collection, while others were mating with them or avoided interaction with them [4]. From observation, younger, inexperienced males were slightly more aggressive toward dummies than older and more experienced individuals. Conducted behavioral research showed that young males more often chase after the female, and this causes her to escape. Female avoidance of overzealous male behavior is beneficial for the female because due to the large sexual dimorphism, the male can potentially kill the female. Such incidents occurred in other breeding centers when during the breeding season one of the males killed one female (personal communication, birds’ keeper). Such situations probably do not occur in the wild due to non-restricted areas, unlike in captivity. It is important in breeding centers and zoological gardens to provide hens safe places without males. In Black grouse (*Tetrao tetrix*), during the selection of males, fighting behavior is evaluated by females over physical characteristics. Aggression and fights are also common between males at the lek area in Capercaillie [18] and related species, Black-billed capercaillie (*Tetrao urogalloides*) [40], but it was not studied yet whether it has the similar function like in Black grouse. Still, we suppose the right direction of aggressive behavior is important, especially knowing that Capercaillie males are predisposed to show aggressive behavior. When antagonistic intersex interactions become more frequent and intense, separation of birds is recommended.

More serious observations of two antagonistic behaviors between female and male not related to breeding showed they were probably caused by competition for food resources. Birds have permanent access to a variety of food, but feeding took place in the morning, so in the afternoon, when both aggressive behaviors were observed (18:49 and 19:49), the quality of leaves and buds was lower. In both cases, the female was more aggressive than the male and tried to peck, while male was only walking toward the female with a spread tail and raised neck. This indicates a normal male response and possible existence of mechanisms mitigating male aggression towards females when male behavioral patterns are not disrupted.

### 4.2. Female/Female Interactions

Studies of Capercaillie aggression are usually made in the context of males at lek areas, while there is no information about female–female antagonistic behavior. It was described that in captivity, hens establish a hierarchy among themselves [41], and while we did not witness female–female aggressive behavior during male tooting activity throughout the year, we have recorded a case where one female chased away another that showed readiness for mating (unpublished data). A similar phenomenon has been observed in Black grouse, when dominant females hampered rivals’ mating [42]. After visiting the lek, females disperse to nest and rear chicks in solitarily. According to Wegge [43], Capercaillie females avoid contact with other females in pre-laying and laying period and Menoni [44] observed signs of territoriality. Those observations are compatible with our studies: females show a series of antagonistic behaviors toward intruders, but only while incubating. Females whose nests were taken by an intruder never showed TA, only PN the intruder from the nest and if there was no success, they may have started AT. Even when the nest was taken over by an intruder, the intruder was the one who was making TA. It appears that the aggression of Capercaillie females is limited to where a female is on a nest, not the fact whose nest it is.

The number of tactics taken by hens to defend and regain the nests was quite various. Birds must confront a variety of situations in the wild, so behavioral plasticity is advantageous. The observed females behaved in various ways: they may abandon the nest after several returns to the occupied nest, tried to push or attack the intruder or even tried to take eggs from the nest and incubated them near the nest. We observed the female that after the first awkward successful attempt to push the intruder from the nest, made the second attempt to push immediately and more firmly, what may suggest that the bird learned that this kind of behavior led to successful usurpation. However, the situations were probably stressful for the females, which may be the reason for aggressive behavior among individuals. We speculate that one of the observed egg-breaking and eating incidents was related to nest site competition. Capercaillie are mainly herbivore and eggs are not a food resource used by this species. All observed behaviors of oviphagy should be treated as pathological behavior. It may be related with the consequences of stress and limited living area [45], calcium deficiency [46,47,48] or after accidental egg breaking learning about a new way of obtaining food [6]. Each case of Capercaillie oviphagy should be considered individually in relation to the individual and environmental conditions.

We suspect that the main cause of female/female antagonistic interactions was related to limited space in the enclosures. Even when females had a relatively large space to set up their nests, competition for nesting places in the breeding center takes place every year, which resulted in nest abandonment or even destruction of eggs. Density of females in the breeding center was one bird per 0.013 hectares while in the wild, the density was one bird per 22.7 hectares [49] and less than two females per 100 hectares [43]. Providing such a large living space for birds in captivity is impossible, so other measures should be taken. To minimize the risk of undesirable effects of nest site competition, large living space for birds with more potentially attractive nesting places and separating individuals with high tendencies to take over nests of other birds should be considered. Nonetheless, the basis for preventing egg loss in captivity due to nesting place competition is a careful observation of birds’ behavior. There was a lack of published information on specific nest site selection by females, and we could not determine what made a specific nesting sites more attractive than others. We recommend more research in this area.

Clutch laying and incubation from multiple individuals is evident from various species [21,50]. Studies have shown that nest sharing is a more common behavior than expected and new reports are emerging for new bird species. Red-breasted nuthatches (*Sitta canadensis*) and Mountain chickadees (*Poecile gambeli*) was described by Robinson et all. [50] as a single observation of successful nesting for both species. Authors speculated that co-nesting was the result of a limited number of potential nesting sites and the absence of a nuthatch male who could chase away a pair of Mountain chickadees. Reports of unusually big grouse clutches were reported from the 80’s [51,52]. Storch and Segelbacher [21], based on the analysis genetic material from unhatched eggs, post hatched eggshells and feathers found an example of nest site adoption, where nest was abandoned and later taken by another individual. When two or more females started to lay eggs in one nest, usually the explanation is intraspecific nest parasitism [53,54,55] or egg adoption [56,57]. Several evidences of egg and chick adoptions were made in Ruffed grouse (*Bonasa umbellus*) [58] species like Capercaillie (subfamily Tetraoninae) and living in a similar habitat and climate zone. Observed interactions in the breeding center indicate nest site competition as both females wanted to incubate. We assume that, in breeding centers, such interactions occurred more often than in the wild due to the restricted living area, but we suspect that they also occur in nature. Joint egg laying and incubation may occur in grouse in the wild rarely [21,51,52], but in captivity, it seems to be a frequent phenomenon, also often correlated with aggression.

## 5. Conclusions

The conducted research showed that, for Capercaillie, the choice of a mate depends on many combined factors: age, tooting activity (latency and daily time of tooting), and behavior toward females. Tooting activity of older (three ant eight years old) males is greater than younger ones (one and two years old). Females spent more time with older males, but the factor of time the female spent with a male cannot be directly related to paternity, so changes of females’ preferences and male activities during the whole breeding season should be taken into consideration. Young males chase after females more often than older ones, and hens avoid contact with males when they become too insistent, so it is important to provide male-free areas for the females to rest.

Nesting site competition in captivity is intense when birds are kept in groups. In most cases, females only assumed a threatening attitude to scare the intruder away from the nest site, but more destructive incidents may occur. Because of nesting place competition, eggs may be destroyed, and nest abandonment may occur. Large living space and an abundance of attractive nesting places (piles of branches arranged in “tents”, wicker baskets, additional bushes) should be available. However behavioral observation should still be made to intervene in cases whereby the competition becomes very intense. Removing non-nesting females from breeding areas should be taken into consideration.

## Figures and Tables

**Figure 1 animals-10-00583-f001:**
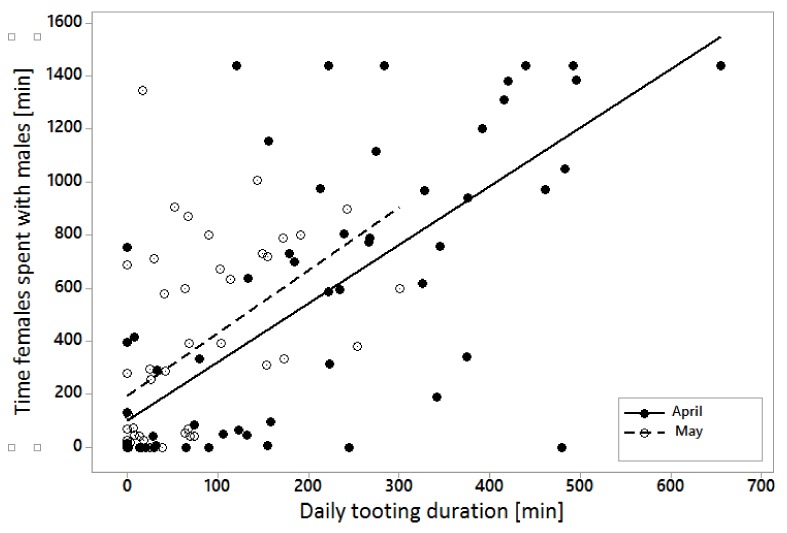
Correlation between Capercaillie males daily tooting duration and time females spent with the males in their sub-units in April and May.

**Figure 2 animals-10-00583-f002:**
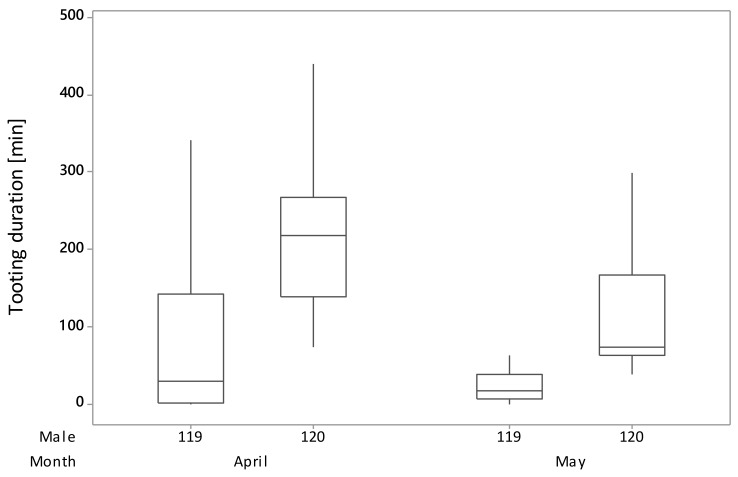
Month-dependent changes of tooting duration in Capercaillie male 119 and 120.

**Figure 3 animals-10-00583-f003:**
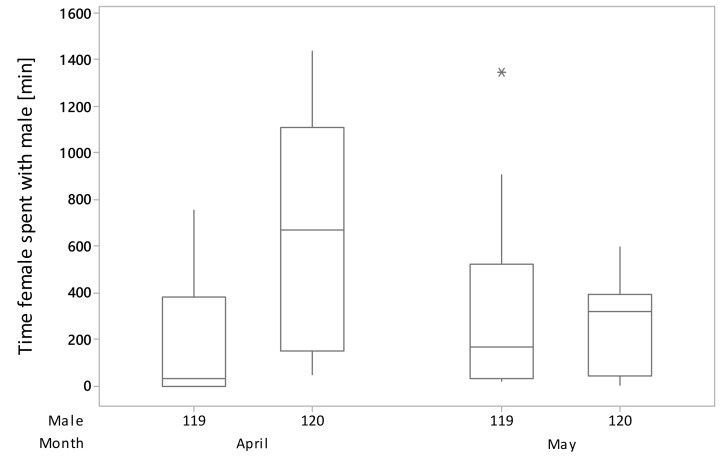
Month-dependent changes in time females spent with male 119 and 120. Outlier observation was marked with “*”.

**Figure 4 animals-10-00583-f004:**
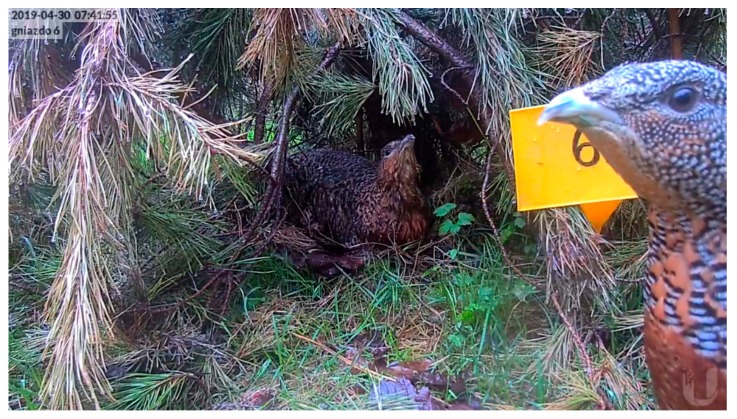
An example of female threatening attitude (TA) with a raised head, ruffled feathers on the throat and making sounds caused by another female presence near the nest.

**Figure 5 animals-10-00583-f005:**
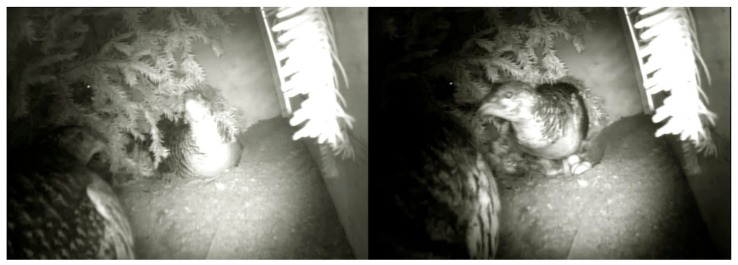
Threatening attitude (TA) of nesting female turns into an attack on the nest intruder (AT).

**Table 1 animals-10-00583-t001:** Ethogram for observed Capercaillie social behaviors in Capercaillie Breeding Centre in Wisła Forestry District.

Type of Social Interactions	Activity	Description
Male activity in presence of female	Tooting	Making the full tooting song with raised, spread tail and lowered wings
	Ignoring	Not paying attention to the female presence, not looking at her direction
	Observing	Observing the female presence but not tooting or walking after female
	Chasing	Chasing after the female
	Mating	Mating with the female
Female activity in presence of male	Ignoring male	Not paying attention to the male presence, not looking at male direction
	Observing male	Observing the male presence but not showing any signs of readiness for mating
	Readiness for mating	Showing readiness for mating by laying to ground and spreading wings
	Avoiding	Avoiding or escaping from male
	Mating	Mating with the male
Female-female interactions	No reaction (NR)	Showing no reaction to the presence of the another female
	Treating attitude (TA)	Showing threatening attitude with a raised head, ruffled feathers on throat and making sounds (Record 1)
	Attack (AT)	Attack by beak one bird to another
	Nest fight (NF)	Fight over nest between two birds by using the beaks
	Pushing (PN)	Attempts of pressing to push a incubating female out from the nest by another female

**Table 2 animals-10-00583-t002:** The set of microsatellite loci used in this study, along with their primer sets, fluorescent dyes and authors.

No.	Locus	Forward/Reverse Primer	Primer Sequence	Dye	Author
1	TUT1	F	GGTCTACATTTGGCTCTGACC	FAM	[28]
		R	ATATGGCATCCCAGCTATGG		
2	TUT2	F	CCGTGTCAAGTTCTCCAAAC	FAM	[28]
		R	TTCAAAGCTGTGTTTCATTAGTTG		
3	TUT3	F	CAGGAGGCCTCAACTAATCACC	CY3	[28]
		R	CGATGCTGGACAGAAGTGAC		
4	TUT4	F	GAGCATCTCCCAGAGTCAGC	HEX	[28]
		R	TGTGAACCAGCAATCTGAGC		
5	BG4	F	ATTCATCAAGTTGGCTTTGGA	FAM	[29]
		R	TCAAGTCTTTTGGGGTGTCATAG		
6	BG6	F	AAAGAGGCAAGCACTCACAATG	HEX	[29]
		R	CCCTTGGAATATCCTTTAACAAAAC		
7	BG10	F	ATGTTTCATGTCTTCTGGAATAG	HEX	[29]
		R	ATTTGGTTAGTAACGCATAAG		
8	BG12	F	TCTCCTTCTAAACCAGTCATTC	CY3	[29]
		R	TAGTTTCCACAGAGCACATTG		
9	BG14	F	ATCCTACTGAACAAAATATCTGC	FAM	[29]
		R	TATGCAGGTAGGTAGTGAGAGAG		
10	BG15	F	AAATATGTTTGCTAGGGCTTAC	FAM	[29]
		R	TACATTTTTCATTGTGGACTTC		
11	BG18	F	CCATAACTTAACTTGCACTTTC	CY3	[29]
		R	CTGATACAAAGATGCCTACAA		
12	BG19	F	CAAGGCGCAACATTAAGATTC	CY3	[29]
		R	TGTATT TTGGAAACTCTGTGTGC		
13	BG20	F	AAGCACTTACAATGGTGAGGAC	FAM	[29]
		R	TATGTTTTCCTTTTCAGTGGTATG		
14	TUD1	F	ATTTGCCAGGAAACTTGCTC	HEX	[28]
		R	AACTACCTGCTTGTTGCTTGG		
15	TUD2	F	GTGACAACTCAGCCCCTGTC	CY3	[28]
		R	AATAAGGGTGCGCATACACC		
16	TUD3	F	TCCAAGGGGAAAATATGTGTG	FAM	[28]
		R	TTCTTCCAGCCCTAGCTTTG		
17	TUD4	F	TTAGCAACCGCAGTGATGTG	HEX	[28]
		R	GGGAGGACTGTGTAGGAGAGC		
18	TUD5	F	CCTTGCTGCACATTTTCTCC	CY3	[28]
		R	GGTGCTGAGCATGTACTAGGG		
19	TUD6	F	GGTGAGCAAGCCACAAATAAC	FAM	[28]
		R	GAGGACTGCAGAACCCACTG		
20	TUD7	F	TGACACTGGGGTCATTAGGC	HEX	[28]
		R	AACATGGGCAGGAGGAGAC		
21	TUD8	F	TGCAGCCTCCTCTAATTTCG	CY3	[28]
		R	CTGGACATCAGCAATCATGC		
22	TTD1	F	AGTGACCTGACAAACCCATC	FAM	[30]
		R	CTCCAAGACAAAGAGAAACTGT		
23	TTD2	F	AACAGCCTGAAATACTGAACTT	HEX	[30]
		R	ATGTGGTTTTGAAGTAAGTTGAC		
24	TTD3	F	CTGAGGGAGCAGTGAATG	CY3	[30]
		R	TCACAGGTGGGCATCTG		
25	TTD4	F	ACATGGTCTTCTTTGCCC	FAM	[30]
		R	AGAACCACTACAGCAGCCTT		
26	TTD5	F	CCTTCCCCCATTCAAAAG	HEX	[30]
		R	GGCTGAAGTTCATTGGCAG		
27	TTD6	F	GGACTGCTTGTGATACTTGCT	CY3	[30]
		R	CATGCAGATGACTTTCAGCA		
28	TTT1	F	GCAGTCCAGCCTTATTTCA	FAM	[30]
		R	TCAGTGCTTCACTAACCTCTT		
29	TTT2	F	GTGAATGGATGGATGTATGAA	HEX	[30]
		R	GTCTGTCAATGAACTTCTTGG		

**Table 3 animals-10-00583-t003:** Month and male age depending tooting latency in Capercaillie Breeding Center in Wisła Forestry District.

Male Number (Age)	April (Tooting Days)	May (Tooting Days)
105 (8)	4:20 (16)	6:08 (12)
120 (3)	4:42 (12)	4:31 (11)
119 (2)	6:56 (16)	7:51 (12)
122 (1)	9:48 (6)	9:41 (1)

**Table 4 animals-10-00583-t004:** Observed antagonistic behavior between Capercaillie females in the nest area during laying and incubation time.

Observed Behavior	Observed Cases	% Of Observed Cases
NR	27	27.27
TA	60	60.61
AT	5	5.05
NF	2	2.02
PN	2	2.02
NR + TA ^1^	2	2.02
TA + AT + PN ^1^	1	1.01

^1^ One female behavior smoothly transitioned into another.

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
