# Peer review of "Female-Male and Female-Female Social Interactions of Captive Kept Capercaillie (Tetrao Urogallus) and Its Consequences in Planning Breeding Programs"

_animals, 2020, doi:10.3390/ani10040583_

Round 1
Reviewer 1 Report
Simple summary
- Line 12: "The capercaillie is..."
- Line 16: "reasons for these problems..."
- Line 17: "research was conducted in the capercaillie breeding centre at..."
- Line 19: higher tooting activity? Please define that this is to the reader.
- Line 25: in captive. No need for captive kept.
Abstract
- Line 28: Suggest: "Female-male and female-female social organisation and social interactions are especially poorly documented".
- Line 35: correlation or relationship?
Introduction
- Line 45: "A main aim..."
- Line 59: "To do this, we must be aware of possible abnormal behaviour performance".
- Line 70: "The capercaillie's secretive..."
- Line 71: "in their natural habitat"
- Line 72: Why?
- Line 72-74: "While there were made several research at lek area [17–19] where birds gather, usually year by year, there is almost nothing know about social interactions outside leks" You need to re-write this sentence. Please get help with English proofreading.
- Line 75: "there readiness to mate"
- Line 78: "some research was made..."
- Line 81-82: Needs re-writing. Does not make sense.
- Line 83: the fact about sensitivity to human disturbance needs a reference.
- Line 90: Needs re-writing. "Simultaneously made research"???? You don't make research. Research is conducted or performed.
Methods
- Line 95: "at the..."
- Line 96-98: The breeding flock consisted annually of 13-15 males and 30-40 females, is kept throughout the year in wooden, roofed aviaries divided into sub-units. Rewrite for clarity
I'm going to stop the review here because the weak sentence structure and poor grammar means this is too much work to read through and edit. This paper needs a thorough re-write. You need to get someone who is fluent in English as a first language to read this paper and re-draft the sentences. You miss out words that would help make sentences flow. The poor level of written English means the paper is hard to follow and the methods are not repeatable.
Please have this paper redrafted and re-written and re-sent for review. That way it will be easier to determine the quality of the science and whether this is a valid or repeatable study. At the moment, it is very difficult to understand what was done and why.
Author Response
Firstly I want to thank you kindly for your review. Below we wrote the explanations and answers. Before we sent to “Animals” journal manuscript we asked for professional language services. Also other two reviewers did not had reservations about the intelligibility of the text. We sent the redactor confirmation of certificate of language correction, and she told us that she will provide you that information. However, we followed the given below comments.
Simple summary
- Line 12: "The capercaillie is..." Corrected
- Line 16: "reasons for these problems..." Corrected
- Line 17: "research was conducted in the capercaillie breeding centre at..." Corrected
- Line 19: higher tooting activity? Please define that this is to the reader. Corrected
- Line 25: in captive. No need for captive kept. Corrected
Abstract
- Line 28: Suggest: "Female-male and female-female social organisation and social interactions are especially poorly documented". Corrected
- Line 35: correlation or relationship? Corrected, it should be relationship
Introduction
- Line 45: "A main aim..." Corrected
- Line 59: "To do this, we must be aware of possible abnormal behaviour performance". Corrected
- Line 70: "The capercaillie's secretive..." Corrected
- Line 71: "in their natural habitat" Corrected
- Line 72: Why? We added explanation: “because of limited availability of observation places and law restrictions that were made for the protection of birds”
- Line 72-74: "While there were made several research at lek area [17–19] where birds gather, usually year by year, there is almost nothing know about social interactions outside leks" You need to re-write this sentence. Please get help with English proofreading. Corrected
- Line 75: "there readiness to mate" Corrected
- Line 78: "some research was made..." Corrected
- Line 81-82: Needs re-writing. Does not make sense. Corrected
- Line 83: the fact about sensitivity to human disturbance needs a reference. References added: Moss, R.; Leckie, F.; Biggins, A.; Poole, T.; Baines, D.; Kortland, K. Impacts of Human Disturbance on Capercaillie Tetrao urogallus Distribution and Demography in Scottish Woodland. Wildlife Biology 2014, 20, 1–18. and Coppes, J.; Ehrlacher, J.; Thiel, D.; Suchant, R.; Braunisch, V. Outdoor recreation causes effective habitat reduction in capercaillie Tetrao urogallus : a major threat for geographically restricted populations. J Avian Biol 2017, 48, 1583–1594.
- Line 90: Needs re-writing. "Simultaneously made research"???? You don't make research. Research is conducted or performed. Corrected
Methods
- Line 95: "at the..." Corrected
- Line 96-98: The breeding flock consisted annually of 13-15 males and 30-40 females, is kept throughout the year in wooden, roofed aviaries divided into sub-units. Rewrite for clarity Corrected

Reviewer 2 Report
The manuscript of Rosenberger et al. deals with an interesting topic by exploring social interactions of captive Capercaillie during breeding season which may provide further notions to prevent population decline. The manuscript is well-written and the results are of practical interest providing new information on animal interaction in this poorly studied species. However, I would like to suggest some minor revisions before publication:
P2, L 50: Please, consider to change ‘what’ with ‘which’. P2, L 55: Please, consider to change ‘One of examples is’ with ‘One example is’. P2, L 62: Please, consider to add ‘studies’ after ‘research’ and to double-check throughout the manuscript because the word ‘research’ is an uncountable noun. For instance in L 86: ‘but still there are no research made’ with ‘ but still there is no research made’ etc. P2, L 62: Please, add ‘this’ between ‘that’ and ‘species’. P2, L 67-69: Please, consider to revise the sentence as follow: ‘Despite research carried out on this topic, little is known about…’ or similar. P2, L 72-74: As for above, please consider to revise the sentence. P2, L 80: Please, consider to change ‘what was’ with ‘as’. P2, L 88-92: The overall aim of the study is not well described by the authors whereby I would suggest to revise the sentence by providing more info. For instance, to explore social interactions during breeding season? or, given the poor literature available in natural environment, to investigate behavioural activity of captive animals for a greater understanding of behavioural patterns in their wild counterparts? P3, L 94-109: I would suggest to include few info about diet and source of water applied within the breeding centre. P3, L 98: Please, add ‘of’ before ‘six small sub-units’. P3, L 127-128: How many hours of video recordings were carried out in total during the whole study and for each recording period (i.e. 1. Recording of male-female interaction from April 15 to May 12; 2. Recording of female activity near the nests during laying phase?) Were observations made on all video recordings or was a sample of hours/per day selected and video analysis was carried out only on this sample? If so, this selection was made according to what? P3, L 126: Please, consider to revise the sentence as follow: ‘..one with different color to allow for identification of individuals…’ or similar P3, L 128: Please consider to include a table with the ethogram of the studied behaviours which I suppose has been used to identify the behavioural interactions observed in the videos. Indeed, not all behaviours displayed by the animals were properly described in paragraph 2.2, for instance in L 122 the authors says ‘recording of females activities..’ – what type of behavioural activity? P3, L 133: Please, add ‘were’ after ‘two people’. P6, L 159: I suppose it should be ‘spent’ instead of ‘spend’. P6, L 187-188: Please, consider to revise the sentence as follow: ‘the relationship between duration of tooting and date of observation was detected’ or similar. P7, L 197-201: Is there any potential environmental factor that may help to explain the reduction of tooting duration of male 120 (which was older than 119 and the one performing longer tooting behaviour in April) between the two months? and in turn females’ choice towards male 119? P8, L 202: Do the authors mean ‘higher’? P8, L 222-238: The authors provide an extensive description of what they classified as ‘aggressive behavior’ performed by the same male and female over 2-day time. What’s the reason for such a detailed description? Is this a behaviour that has never been described before in captivity and/or nature whereby justifying this deep explanation? Moreover, the cases observed were only 2 over c. a month of video recordings, what’s the possible implication for this finding? – A similar detailed description is provided for female/female interactions in L 255-271. P12, L 377-378: Did the authors check, for instance, any correlations between density and antagonistic interactions to provide a more robust evidence for justifying their assumption?Author Response
Firstly I want to thank you kindly for your review. Below we wrote the explanations and answers:
P2, L 50: Please, consider to change ‘what’ with ‘which’. Done
P2, L 55: Please, consider to change ‘One of examples is’ with ‘One example is’. Done
P2, L 62: Please, consider to add ‘studies’ after ‘research’ and to double-check throughout the manuscript because the word ‘research’ is an uncountable noun. For instance in L 86: ‘but still there are no research made’ with ‘ but still there is no research made’ etc. Done
P2, L 62: Please, add ‘this’ between ‘that’ and ‘species’. Done
P2, L 67-69: Please, consider to revise the sentence as follow: ‘Despite research carried out on this topic, little is known about…’ or similar. Done
P2, L 72-74: As for above, please consider to revise the sentence. Done
P2, L 80: Please, consider to change ‘what was’ with ‘as’. Done
P2, L 88-92: The overall aim of the study is not well described by the authors whereby I would suggest to revise the sentence by providing more info. For instance, to explore social interactions during breeding season? or, given the poor literature available in natural environment, to investigate behavioural activity of captive animals for a greater understanding of behavioural patterns in their wild counterparts? Done. We improved paragraph and added “. Made observations may improve our understanding of Capercaillie breading behavior in captive kept individuals, but it may also improve our knowledge of behavioral patterns of their wild counterparts.”
P3, L 94-109: I would suggest to include few info about diet and source of water applied within the breeding centre. Done. I added information about providing food and water at the end of paragraph 2.1 (Maintenance conditions)
P3, L 98: Please, add ‘of’ before ‘six small sub-units’. Done
P3, L 127-128: How many hours of video recordings were carried out in total during the whole study and for each recording period (i.e. 1. Recording of male-female interaction from April 15 to May 12; 2. Recording of female activity near the nests during laying phase?) Were observations made on all video recordings or was a sample of hours/per day selected and video analysis was carried out only on this sample? If so, this selection was made according to what? We added information. All records, 24h a day, for males were analyzed (that takes 648 h per male). In case of females nests were observed all the time, 24 h/day, but records were only when saved automatically when movement in front of camera was detected. Nests records were analyzed daily in case when any intervention should be made. Male records were stored and analyzed after breeding season.
P3, L 126: Please, consider to revise the sentence as follow: ‘..one with different color to allow for identification of individuals…’ or similar Done
P3, L 128: Please consider to include a table with the ethogram of the studied behaviours which I suppose has been used to identify the behavioural interactions observed in the videos. Indeed, not all behaviours displayed by the animals were properly described in paragraph 2.2, for instance in L 122 the authors says ‘recording of females activities..’ – what type of behavioural activity? We added proposed ethogram in paragraph 2.2 Making observations
P3, L 133: Please, add ‘were’ after ‘two people’. Done
P6, L 159: I suppose it should be ‘spent’ instead of ‘spend’. Done
P6, L 187-188: Please, consider to revise the sentence as follow: ‘the relationship between duration of tooting and date of observation was detected’ or similar. Done
P7, L 197-201: Is there any potential environmental factor that may help to explain the reduction of tooting duration of male 120 (which was older than 119 and the one performing longer tooting behaviour in April) between the two months? and in turn females’ choice towards male 119? According to us there was any environmental factor that may affect bird behavior. Both birds were feed the same, had the same living space, the same temperature and duration of daylight. We suspect that was individual factor (maybe level of hormones) or maybe the fact that were able to hear other males, what may affected their tooting activity. Unfortunately we were not able to check this.
P8, L 202: Do the authors mean ‘h her’? I do not understand question.
P8, L 222-238: The authors provide an extensive description of what they classified as ‘aggressive behavior’ performed by the same male and female over 2-day time. What’s the reason for such a detailed description? Is this a behaviour that has never been described before in captivity and/or nature whereby justifying this deep explanation? Moreover, the cases observed were only 2 over c. a month of video recordings, what’s the possible implication for this finding? – A similar detailed description is provided for female/female interactions in L 255-271. As far we know, any descriptions of Capercaillie aggressive in captivity were not made. We decided to describe them in so detailed way because we know that in captivity as consequence of this aggression behavior females may be killed. That was reported from birds keepers, but never was seen how aggression between male and female looked like. We think that competition for food may be one of that reasons (what we suspect after video analyze). Even that it may not seems frequent, we think that detailed descriptions may be helpful, especially for bird keepers when they will have problems with bird aggression. The same reason was why we described with so many details female/female interaction.
P12, L 377-378: Did the authors check, for instance, any correlations between density and antagonistic interactions to provide a more robust evidence for justifying their assumption? It is only our assumptions and the observations conducted by bird keepers. For years breeding centers avoid to put more than 4-5 females into one family so we were not able to investigate how density affected nesting-site competition.

Reviewer 3 Report
This study has the potential to improve the management of captive Capercaillie species. That said, a few large questions/issues lingered after I reviewed the manuscript.
- It is difficult for me to evaluate the significance of this research and interest to readers. Although these birds are endangered in the wild, what is there status in captivity? Will there be a large number of birds that benefit from the results of this study?
- These birds are clearly difficult to observe in the wild, but I would like to see more information about their behavior, especially a detailed ethogram.
- It is unclear to me why paternity tests are relevant. The title of the study suggests a descriptive study about behavior, but the conclusions seem to point more to mate choice. If this is a study about mate choice, and not simply about interactions between sexes, then it makes more sense.
- Please define how you used "latency" in the study. What is "tooting latency" (page 6 of 17)?
- Please clarify what it means in the Results section that "older males were starting their activity earlier." Earlier in the day? Night?
- A chart or two that clearly display the results of the study, to compliment the text, would be appreciated.
- If we are unable to provide larger spaces for the birds, what other measures should be taken, as you suggest on page 12 of 17? Intervention?
- I would like to see a tighter relationship between the results and the discussion. As I stated earlier, the title, results, and conclusions don't entirely mesh to me.
Author Response
Firstly I want to thank you kindly for your review. Below we wrote the explanations and answers:
1. It is difficult for me to evaluate the significance of this research and interest to readers. Although these birds are endangered in the wild, what is there status in captivity? Will there be a large number of birds that benefit from the results of this study?
In Poland are at three, big breeding centres (Wisła Foresty, Leżajsk Forestry, Głęboki Bród Forestry), we know about others in Czech Republic, Spain and Germany. Breeding centre in Wisła Forestry often have guests from other countries that came to learn more about breeding Capercaillie in captivity. Capercaillies are also kept in some zoological gardens.
2. These birds are clearly difficult to observe in the wild, but I would like to see more information about their behavior, especially a detailed ethogram.
We added ethogram of observed behaviours.
3. It is unclear to me why paternity tests are relevant. The title of the study suggests a descriptive study about behavior, but the conclusions seem to point more to mate choice. If this is a study about mate choice, and not simply about interactions between sexes, then it makes more sense.
It was believed (at least it was considered among people who deal with the breeding of Capercaillies in captvity), that older males are preferred by females. We wanted to check does age factor will take major role in mate selection. We also had a question, can relaying only on female interest toward males (i.e time spent with males), can we suppose who will be the father of offspring. Genetic test was made to be 100% certain of who was the father.
4. Please define how you used "latency" in the study. What is "tooting latency" (page 6 of 17)?
Explained in paragraph “2.2 Making observations”
5. Please clarify what it means in the Results section that "older males were starting their activity earlier." Earlier in the day? Night?
We added explanation (in first paragraph of “results”) and table with hours when tooting latency is show.
6. A chart or two that clearly display the results of the study, to compliment the text, would be appreciated.
In results section we added one table (with month and male age depending tooting latency in Capercaillie Breeding Center in Wisła Forestry District) and chart (with correlation between capercaillie males daily tooting duration and time females spent with the males in their sub-units.).
7. If we are unable to provide larger spaces for the birds, what other measures should be taken, as you suggest on page 12 of 17? Intervention?
We addend our propositions at the end of the conclusion paragraph. Removing “problematic” females may be one of resolutions. Other is providing more potentially attractive nesting places. In any case increasing bird density in the enclosure (as a result of the desire to increase the number of offspring) should not be used, because it causes high nesting places competition, potential stress, and does not increase the number of obtained offspring at all.
7. I would like to see a tighter relationship between the results and the discussion. As I stated earlier, the title, results, and conclusions don't entirely mesh to me.
We changed a title, expanded part of discussion and conclusions.

Round 2
Reviewer 1 Report
There are improvements to this manuscript that makes it more readable and easier to follow.
The subject of this manuscript is a relevant one and it's great to see work on such endangered species that can be overlooked.
You still need to work on clarity of written English.
Further points for consideration.
Title
Suggest: An evaluation of within and between sex social interactions in capercaillie: implications for conservation breeding initiatives
Introduction
Line 64: One example...
Line 70: "behavioural disorders"
Line 72: Not really pathological. "Published research provides evidence for abnormal behaviour performance in captive capercaille..."
Line 81: "The capercaille's..."
Line 84: legal restriction.
Line 101: good study population
Line 102: but there is still no published reason on...
Line 107: Suggest "Observations on captive capercaille may improve our understanding of breeding behavior in a managed environment, and may also improve our knowledge of behavioral patterns of their wild counterparts.
Provide a reason why wild counterparts may also benefit.
E.g. ". By providing evidence of high-value habitat features (essential for the perfomance of breeding behaviour) that should be conserved.
Line 109: Suggest "Research effort is essential to understanding the impact of the conditions Capercaillie experience in breeding centers and zoological gardens, and this providing evidence for potential improvements, as well as avoiding factors that may reduce breeding success."
Methods
Line 118: "The aviary where..."
Line 122: What do you mean "screened daily"? Do you mean cleaned? Raked over? Turned? Kept hygienic?
Line 123: Poured? Do you mean "with a sand flooring"?
Line 124: "...that was partly forested..."
Line 125: What are windthrows?
Line 125: Re-write as "Females were provided with XXm2 of space and males XXm2 of space within their aviaries".
Line 127: "The study flock was..." "...that consisted of..."
Line 130: "thanks to holes (diameter 17-18cm) within the adjoining sides of the enclosure..."
Line 134: "... were an additional source of food."
Line 134: "Water (acidified with lemon juice) was..." And give the reason why acidification was needed.
Line 138: Suggest "Three remote cameras were used for accurate recording of the birds' behavior in each sub-unit. The rest of the cameras (one camera per nest) were placed near the nests to monitor behavior during incubation."
Line 141: "For this research..."
Line 144: Sog? Or song?
Line 146: "... took a "tooting attitude" with..."
Line 147: Suggest "Interactions between the sexes were categorized into four types. For male: 1. Ignore female presence; 2. Observe female presence; 3. Tooting; 4. Chasing female. For female: 1. Ignore male presence; 2. Observe male; 3. Interested in male; 4. Escape from male."
Line 151: "... female-female interactions..." Is this what you mean? and "in the nest..."
Line 152: "...was made..." "... during the..."
Line 153: "the bird's keepers..."
Line 154: "and continued for..."
Line 155: "... wore two types..."
Line 157: Provide information on how birds were accustomed to the cameras.
Line 159: "near a female's nest..." "were motion activated to start recording when a bird was present to save..."
Ethogram: edits below.
Not looking in her direction
Observing the female's presence
Not paying attention to the male's presence
not looking in the male's direction
* Please check all the way through... "female's direction or male's direction or female's presence or male's presence" *
"... by remaining close to the ground..."
"...and making sounds..." What sounds?
Attacking another bird, using the beak
...by using their beaks
Attempted pushing of a female off her nest by another female.
Line 166: "...with the mating process" "... we used..."
Line 168: "for every adult and juvenile kept at the breeding centre..."
Line 184: alleles?
Line 186: "with all loci"
Line 192: Start this section with "Data were analysed using... give programme name and version."
Line 192: "Data were not normally distributed (give the test used to determine normality)..."
Line 193: Suggest "Any relationship between daily time that females spent with males in their sub-units and tooting duration, as well as any correlation between female presence in male sub-unit at night and tooting duration were analysed using a two-tailed Wilcoxon test"
Line 195: What do you mean male's factor?
Line 196: "any influence of date"
Line 198: Does not make sense. Please rewrite
Line 198: "using a Kruskal..."
Line 201: "A Pearson..." "...any relationship..."
Line 202: "... conducted at the 95% confidence level."
What alpha level for significance? 5%?
Line 209: "Dr" and what do you mean "for keeping"?
Results
* all the way through the paper, when you refer to specific individual in your sample population I suggest you give it a capital for clarity (e.g. Male 105). *
Line 218: Again, male factor is still confusing.
Line 223: for spending time...
Line 225: was found between...
Line 233: was found.
Line 234: male 105's...
Line 237: neither when analyzed data...
Line 239: Suggest "To further investigate why..."
Line 246: with this male?
252: "...were correlated X2..."
Line 257" Suggest "Male 122 was most often..."
I cannot see any of the figures to review. They are just blank squares.
Line 276: "between a female and..."
Line 288: "The female stood up..."
Line 294: Suggest "During four years of observation (tracing the fate of 18 nests), we witnessed 99 interactions in distance about 50 cm from the nest (for 12 of these nests)."
Line 296: female interactions
Line 296 to end of section: Please re-structure and make more concise. This is hard to follow and full of discursive information that reads more like discussion than results section. The figures are useful though.
Discussion
The discussion is very long. And talks around the subject. Please re-write the discussion to focus on what your results mean. Put your results into the context of what your birds did and why. Then talk about conservation and planning for conservation action. And then present on how your results provide evidence for management. Think about telling a more concise and more succinct story of your findings so that you give a wider application of what you have found.
Author Response
Once again we would like to thank you for review. Below are our answers and explanations.
Title
Suggest: An evaluation of within and between sex social interactions in capercaillie: implications for conservation breeding initiatives Changed as suggested.
Introduction
Line 64: One example... Done
Line 70: "behavioural disorders" Journal allow to use both American and British English, we choosed American
Line 72: Not really pathological. "Published research provides evidence for abnormal behaviour performance in captive capercaille..." Changed as suggested
Line 81: "The capercaille's..." Done
Line 84: legal restriction. Done
Line 101: good study population Done
Line 102: but there is still no published reason on... Changed as suggested
Line 107: Suggest "Observations on captive capercaille may improve our understanding of breeding behavior in a managed environment, and may also improve our knowledge of behavioral patterns of their wild counterparts.
Provide a reason why wild counterparts may also benefit. E.g. ". By providing evidence of high-value habitat features (essential for the perfomance of breeding behaviour) that should be conserved.
Sentence changed as suggested and added reason: understanding female mate choice, and therefore the organization of the leks and their evolution in lekking species. We did not think that reference to habitat features in context of conduced research would be the best choice.
Line 109: Suggest "Research effort is essential to understanding the impact of the conditions Capercaillie experience in breeding centers and zoological gardens, and this providing evidence for potential improvements, as well as avoiding factors that may reduce breeding success." Sentence changed as suggested
Methods
Line 118: "The aviary where..." Done
Line 122: What do you mean "screened daily"? Do you mean cleaned? Raked over? Turned? Kept hygienic? Explained in text: in order to remove all droppings and food remains
Line 123: Poured? Do you mean "with a sand flooring"? Corrected: “Beside small sub-units available for birds for the entire year, aviaries had six small outside areas 4x4.5 m and a height of 4 m where the ground also was covered by sand and four large fenced yards…”
Line 124: "...that was partly forested..." Done
Line 125: What are windthrows? Trees damaged, uprooted or broken by wind.
Line 125: Re-write as "Females were provided with XXm2 of space and males XXm2 of space within their aviaries". Done
Line 127: "The study flock was..." "...that consisted of..." Done
Line 130: "thanks to holes (diameter 17-18cm) within the adjoining sides of the enclosure..." Done
Line 134: "... were an additional source of food." Done
Line 134: "Water (acidified with lemon juice) was..." And give the reason why acidification was needed. Done. Acidification in made to prevent excessive development of gastrointestinal microbiota in the cecum, and it consequence, inflammation. In natural environment Capercaillie live in forests with humic acids in the water.
Line 138: Suggest "Three remote cameras were used for accurate recording of the birds' behavior in each sub-unit. The rest of the cameras (one camera per nest) were placed near the nests to monitor behavior during incubation." Done
Line 141: "For this research..." Done
Line 144: Sog? Or song? Corrected
Line 146: "... took a "tooting attitude" with..." Corrected
Line 147: Suggest "Interactions between the sexes were categorized into four types. For male: 1. Ignore female presence; 2. Observe female presence; 3. Tooting; 4. Chasing female. For female: 1. Ignore male presence; 2. Observe male; 3. Interested in male; 4. Escape from male." Done as suggested
Line 151: "... female-female interactions..." Is this what you mean? and "in the nest..." Clarified, directly in the nest and close to the nest area, up to 50 cm from nests.
Line 152: "...was made..." "... during the..." Corrected
Line 153: "the bird's keepers..." Corrected
Line 154: "and continued for..." Corrected
Line 155: "... wore two types..." Corrected as suggested
Line 157: Provide information on how birds were accustomed to the cameras. Done as suggested. Birds saw cameras before breeding season started.
Line 159: "near a female's nest..." "were motion activated to start recording when a bird was present to save..." Done as suggested
Ethogram: edits below.
Not looking in her direction; Observing the female's presence; Not paying attention to the male's presence; not looking in the male's direction/ * Please check all the way through... "female's direction or male's direction or female's presence or male's presence" * Corrected
"... by remaining close to the ground..." Changed
"...and making sounds..." What sounds? It is hard to describe, it was not exact call, nor alarm call, little similar but the sounds were longer (not as any of records that can be found at https://www.xeno-canto.org/species/Tetrao-urogallus?dir=0&order=typ&pg=1) We will propose adding record to the manuscript.
Attacking another bird, using the beak Corrected
...by using their beaks Corrected
Attempted pushing of a female off her nest by another female. Corrected
Line 166: "...with the mating process" "... we used..." Done
Line 168: "for every adult and juvenile kept at the breeding centre..." Done
Line 184: alleles? Yes, done
Line 186: "with all loci" Done
Line 192: Start this section with "Data were analysed using... give programme name and version." Done
Line 192: "Data were not normally distributed (give the test used to determine normality)..." Done, used t-distribution
Line 193: Suggest "Any relationship between daily time that females spent with males in their sub-units and tooting duration, as well as any correlation between female presence in male sub-unit at night and tooting duration were analysed using a two-tailed Wilcoxon test" Done as suggested
Line 196: "any influence of date" Done
Line 198: Does not make sense. Please rewrite. Done, rewritten. Kruskal-Wallis test was used to test particular male factor’s influence on tooting latency, tooting duration and female presence; any influence of date on tooting latency, tooting duration and duration of female spent with the male; time females spent with the male and their readiness for mating.
Line 198: "using a Kruskal..." Done
Line 201: "A Pearson..." "...any relationship..." Done
Line 202: "... conducted at the 95% confidence level." Done
What alpha level for significance? 5%? Yes, we added that
Line 209: "Dr" and what do you mean "for keeping"? Dr -> PhD (changed), "for keeping" – deleted, it was for keeping birds in captivity, however it was mentioned before.
Results
Line 223: for spending time... Done
Line 225: was found between... Done
Line 233: was found. Done
Line 234: male 105's... Changed
Line 237: neither when analyzed data... Done
Line 239: Suggest "To further investigate why..." Changed as suggested
Line 246: with this male? Done. Yes, male 129.
252: "...were correlated X2..." Done
Line 257" Suggest "Male 122 was most often..." Sentence changed
Line 276: "between a female and..." Done
Line 288: "The female stood up..." Done
Line 294: Suggest "During four years of observation (tracing the fate of 18 nests), we witnessed 99 interactions in distance about 50 cm from the nest (for 12 of these nests)." Done
Line 296: female interactions Done
Line 296 to end of section: Please re-structure and make more concise. This is hard to follow and full of discursive information that reads more like discussion than results section. The figures are useful though.
Rewritten. Also some explanation about presentation of results in section 3.2 and 3.3. The high descriptiveness of the relationship between females and females/males was made in purpose. As far we know, any descriptions of Capercaillie aggressive in captivity were not made. For example of male/female aggression we decided to describe it in so detailed way because we know that in captivity as consequence of this aggression behavior females may be killed. That was reported from birds keepers, but never was seen how aggression between male and female looked like. We think that competition for food may be one of that reasons (what we suspect after video analyze). Even that it may not seems frequent, we think that detailed descriptions may be helpful, especially for bird keepers when they will have problems with bird aggression. The same reason was why we described with so many details female/female interaction. While they may seem incidental, when we look that is occurred every year, as far we know, in other breeding centers, it might reducing welfare and breeding success there.
Discussion
The discussion is very long. And talks around the subject. Please re-write the discussion to focus on what your results mean. Put your results into the context of what your birds did and why. Then talk about conservation and planning for conservation action. And then present on how your results provide evidence for management. Think about telling a more concise and more succinct story of your findings so that you give a wider application of what you have found.
We are aware that the discussion part is very long, but to describe our results, that were not clearly yes/no, we needed to do that way to not lost the background that we think is important. We also arranged the discussion according to the cause and effect chain:
- Mating system in birds and the place of Capercaillie in that context.
- Capercaillie mate selection in the wild and captivity – differences in birds behavior. By what those differences might be caused?
- Mate selection in captivity: role male age factor, tooting display. Can we foresee who was will be the father looking at birds, males and females behavior? Because results were not clear we tried to answer why and does changes in tooting activity might be involved. Here, in new version we expanded the part of tooting song quality factor.
- Next we moved to male/female aggression problem. We started with a description of our pervious observations – aggression toward female dummies, behavior of males depending from their age and reaction of females when they are chased. We added here observations that were made on black grouse, species that is closely related to Capercaillie. We also separated the aggression that was seen during display and the aggression observed during the competition for food, because of various causes of aggression. Once again at the end of this part, we made conclusions.
In next section, female-female interactions, we started with short introduction of described in literature aggression between hens, that were investigated mostly at leks. Because that was not the main subject of our studies, after that introduction we moved to that little data from literature that describe interactions between hens during nesting. Our observations confirms signs of territoriality of nesting females. We discuss the variation of female reactions that occurred during those interactions and we were surprised that nest site competition had so various consequences, many different females antagonistic reactions, attempts of taking eggs from the nest, nest abandonment and egg eating. We suppose all of those is related to higher density in captivity compared to that in natural environment. Providing so much living space in captivity is impossible, so we propose close observation of the females. We end this paragraph with information that even that nest site competition and joint laying occurred in captivity, this also happens in the wild, in many different bird species and in Capercaillie too.
In conclusions we gathered all most important information and provide tips for keeping and breeding Capercaillie.
Reviewer 3 Report
This manuscript is much improved. The authors clearly took the reviewers' comments seriously, with an intention of providing a manuscript suitable for publication. I have just a few comments:
The change in title better reflects the content of the manuscript. Might I suggest something like: Female-male and female-female social interactions of Capercaillie (Tetrao urogallus): implications for captive breeding programs.
Line 35: add "tooting" before "duration"
Line 46: I suspect you mean "progeny"
Line 114: happy to see the addition of more husbandry information
Line 130: female-female interactions
Table 1: the ethogram is a welcome addition, but a bit unclear. There appears to be two sections that outline activity of males in the presence of females. Is one supposed to be for female activity in the presence of male?
Line 188: Figure abc?
Line 199: this sentence could be a bit clearer (i.e., the relationship between duration of tooting...)
Line 452: instead of "However still", perhaps consider "Further".
Author Response
Once again we would like to thank you for review. Below are our answers and explanations.
The change in title better reflects the content of the manuscript. Might I suggest something like: Female-male and female-female social interactions of Capercaillie (Tetrao urogallus): implications for captive breeding programs. We had changed title to “An evaluation of within and between sex social inseractions in Capercaillie (Tetrao urogallus): implications for conservation breeding initiatives” as closer to suggested by all reviewers
Line 35: add "tooting" before "duration" done
Line 46: I suspect you mean "progeny" yes, we corrected it
Line 114: happy to see the addition of more husbandry information He added more information about husbandry: ass information that yard is only available for females, also all sentences: “For each family group three small sub-units were available: left and right only for one of the males, middle only for the females. Atmospheric and lighting conditions were identical for both family groups, and both outside aviaries faced south, which provided plentiful sunshine.” and “Birds were fed once a day with poultry mash, live crickets, fruits (blueberries, cranberries) and fresh buds of deciduous trees. Needles from pine and spruce trees placed inside aviaries were additional source of food. Water acidified with lemon juice was changed daily. To reduce risk of disease, food and water vessels were daily washed and sterilized at 200 0C. To prevent birds from eating food that might have been contaminated during contact with the ground, nourishment was placed in bowls that were hanged up above boxes with a mesh.”
Line 130: female-female interactions done
Table 1: the ethogram is a welcome addition, but a bit unclear. There appears to be two sections that outline activity of males in the presence of females. Is one supposed to be for female activity in the presence of male? Yes, thank you, it should be: Female activity in presence of male.
Line 188: Figure abc? Sorry and thank you for noticing that. It should be figure 1.
Line 199: this sentence could be a bit clearer (i.e., the relationship between duration of tooting...) done
Line 452: instead of "However still", perhaps consider "Further". done